# Microbiota from Preterm Infants Who Develop Necrotizing Enterocolitis Drives the Neurodevelopment Impairment in a Humanized Mouse Model

**DOI:** 10.3390/microorganisms11051131

**Published:** 2023-04-26

**Authors:** Jing Lu, Alexander Drobyshevsky, Lei Lu, Yueyue Yu, Michael S. Caplan, Erika C. Claud

**Affiliations:** 1Department of Pediatrics, Division of Biological Sciences, Pritzker School of Medicine, University of Chicago, Chicago, IL 60637, USA; 2Department of Pediatrics, NorthShore University HealthSystem, Evanston, IL 60202, USA

**Keywords:** necrotizing enterocolitis, microbiota, neurodevelopment

## Abstract

Necrotizing enterocolitis (NEC) is the leading basis for gastrointestinal morbidity and poses a significant risk for neurodevelopmental impairment (NDI) in preterm infants. Aberrant bacterial colonization preceding NEC contributes to the pathogenesis of NEC, and we have demonstrated that immature microbiota in preterm infants negatively impacts neurodevelopment and neurological outcomes. In this study, we tested the hypothesis that microbial communities before the onset of NEC drive NDI. Using our humanized gnotobiotic model in which human infant microbial samples were gavaged to pregnant germ-free C57BL/6J dams, we compared the effects of the microbiota from preterm infants who went on to develop NEC (MNEC) to the microbiota from healthy term infants (MTERM) on brain development and neurological outcomes in offspring mice. Immunohistochemical studies demonstrated that MNEC mice had significantly decreased occludin and ZO-1 expression compared to MTERM mice and increased ileal inflammation marked by the increased nuclear phospho-p65 of NFκB expression, revealing that microbial communities from patients who developed NEC had a negative effect on ileal barrier development and homeostasis. In open field and elevated plus maze tests, MNEC mice had worse mobility and were more anxious than MTERM mice. In cued fear conditioning tests, MNEC mice had worse contextual memory than MTERM mice. MRI revealed that MNEC mice had decreased myelination in major white and grey matter structures and lower fractional anisotropy values in white matter areas, demonstrating delayed brain maturation and organization. MNEC also altered the metabolic profiles, especially carnitine, phosphocholine, and bile acid analogs in the brain. Our data demonstrated numerous significant differences in gut maturity, brain metabolic profiles, brain maturation and organization, and behaviors between MTERM and MNEC mice. Our study suggests that the microbiome before the onset of NEC has negative impacts on brain development and neurological outcomes and can be a prospective target to improve long-term developmental outcomes.

## 1. Introduction

It is estimated that approximately 40% of neonatal necrotizing enterocolitis (NEC) patients develop neurodevelopmental impairment (NDI) [1,2,3,4,5,6,7,8,9,10,11,12,13,14]. The most prevalent NDIs in the NEC population appear to be cerebral palsy (20% of neonates with NEC), cognitive impairment (36%), and psychomotor impairment (35%). Understanding the mechanisms for the development of these complications in infants with NEC is necessary to potentially prevent NEC or to provide treatments to warrant better long-term outcomes. The etiology of NEC remains elusive. It is proposed that prematurity, especially the immaturity of the gastrointestinal tract and immune system, predisposes preterm infants to develop NEC [15,16,17,18,19]. The intestinal epithelium represents the first line of defense against invading organisms to resolve local and systemic inflammation; consequently, aberrant bacterial colonization is considered to be one of the main contributing factors for the pathogenesis of NEC [20,21].

There are distinct characteristics of the early postnatal assembly of microbiota associated with the development of NEC [22]. Our previous studies demonstrated that microbiota samples from preterm infants before the onset of NEC clustered distinctly from those of non-NEC controls and were signified by reduced overall microbial diversity and an expansion of *Gammaproteobacteria* species [23,24]. A meta-analysis study confirmed that microbial dysbiosis characterized by an increased relative abundances of *Proteobacteria* and a decreased relative abundances of *Firmicutes* and *Bacteroidetes* predispose preterm infants to NEC [25]. In a prospective case-control study [26], a relative abundance of *Gammaproteobacteria* and a relative rareness of *Negativicutes* preceded NEC development in preterm infants when compared with gestational age (GA) and birth weight-matched controls. In rodent models of NEC, colonization with two genus, *Serratia marcescens* and *Klebsiella pneumoniae,* from the *Gammaproteobacteria* class is a prerequisite for the development of NEC [19]. These studies underlined the notion that aberrant bacterial colonization preceding NEC contributes to the pathogenesis of NEC.

Gut microbiota is essential for normal gut physiology and contributes to the development of brain and host behavior through appropriate communication along the gut–brain axis [27,28,29,30,31,32,33,34,35,36,37,38]. Conversely, intestinal dysbiosis can adversely influence gut physiology, gut–brain axis signaling, and central nervous system (CNS) functions. Based on our previous study using a humanized gnotobiotic model in which we gavaged preterm fecal samples to pregnant germ-free (GF) mice, preterm microbial communities associated with poor overall postnatal growth adversely affected villus height, crypt depth, enterocyte proliferation, and the number of goblet cells and Paneth cells in the offspring [39]. These preterm microbial communities also resulted in delayed neuronal and oligodendrocyte development and increased systemic and neuroinflammation [40,41]. We further demonstrated that maturation of preterm microbiota, defined by advancing post menstrual age (PMA, defined as GA + weeks of life of donor samples collected and actually used to colonize the mice), PMA-dependent bacterial diversity, dominance of key taxa, and microbial metabolites, predicts better cognitive functions in adult mice [42]. Our findings demonstrated that immaturity of the preterm infant microbiota had significant impacts on small intestine development, near postnatal brain development, and neurological outcomes, suggesting a critical window of early optimization to affect maturation of the preterm infant microbiota and ultimately improve neurodevelopmental outcomes.

Whether the intestinal microbiota in preterm infants who went on to develop NEC has an impact on CNS development and functions is not known. In this study, we investigated the gut microbiota effects on neurodevelopmental outcomes by transfaunating human fecal samples from preterm infants who were diagnosed with stage II B NEC later in life and from healthy term infants to GF dams. We demonstrated specific gut microbiota composition and brain metabolic characteristics, highlighting the differences between the two microbial communities. Most importantly, in the offspring juvenile mice colonized with early microbial communities from NEC patients, we observed impaired tight junction development, intestinal inflammation, delayed maturation of neurons and myelination of oligodendrocytes, deficits in white matter organization and locomotion, and memory defects.

## 2. Materials and Methods

### 2.1. Clinical Study Design

Patient samples were obtained from the Center for the Science of Early Trajectories (SET) biorepository from The University of Chicago Comer Children’s Hospital, a level IV NICU in Chicago, Illinois, from 2015 to 2022. The study was approved by the Institutional Review Board. All methods were performed in accordance with the relevant guidelines and regulations. Consent for the study was obtained from the infants’ parent(s). All infants enrolled were delivered between 24 and 40 weeks of gestation age (GA) and had no congenital anomalies.

### 2.2. Animals

Animal care and experimental procedures were approved by Institutional Animal Care and Use Committees of The University of Chicago (protocol number #71703) and NorthShore University HealthSystem (protocol number EH16-264), strictly in accordance with all guidelines from the U.S. National Institutes of Health. GF C57/BL6J mice were maintained in the gnotobiotic facility of the Digestive Disease Research Core Center at the University of Chicago. All groups of mice were allowed ad libitum access to NIH-31 GF chow and water.

### 2.3. Preparation of Human Fecal Samples for Transfaunation to GF Mice

Transfaunation protocols were carried out as previously described [41]. Aliquots of frozen fecal samples from two patients who later developed Stage II NEC (100 mg per patient) with GAs of 28 weeks (MNEC) and from two full term healthy infants at GA of 39 weeks and 40 weeks (MTERM) were resuspended under anaerobic conditions in 5 mL phosphate-buffered solution (PBS). Homogenates were clarified by 100 μM pore-diameter nylon filters (BD Falcon) before being stored in PBS containing glycerol (final concentration 15% *v*/*v*) at −80 °C in 600 μL aliquots. To initiate microbial colonization, timed-pregnant GF eight- to nine-week-old mice (estimated between E15-17) were gavaged with 0.25 mL aliquots of fecal supernatant. Pups were delivered spontaneously, and litters remained with the mother until weaning.

### 2.4. 16S rRNA Sequencing

Mouse fecal samples were submitted to the Microbiome Metagenomics Facility (MMF) of the Duchossois Family Institute (DFI) at the University of Chicago for genomic DNA extraction and subsequent 16S rRNA gene sequencing on the Illumina MiSeq platform, as previously described [43]. Dada2 (v1.18.0) was used for processing MiSeq 16S rRNA reads with minor modifications in R (v4.0.3) [44,45]. Specifically, reads were first trimmed at 190 bp for both forward and reverse reads to remove low quality nucleotides. Chimeras were detected and removed using the default consensus method in the dada2 pipeline. Then, ASVs with lengths between 320 bp and 365 bp were kept and deemed as high quality ASVs. The taxonomy of the resultant amplicon sequence variants (ASV)s were assigned to the genus level using the RDP classifier (v2.13) with a minimum bootstrap confidence score of 80. Species-level classification can be provided using blastn (v2.13.0) and the refseq_rna database (updated 10 June 2022). Multivariate analysis of the Aitchison distance matrices of microbial composition was additionally employed using PERMANOVA (permutational MANOVA) through R package vegan.

### 2.5. Immunohistochemistry

Ileum segments from two-week-old pups were fixed in buffered formalin and then paraffin embedded for sectioning. Amounts of 5 μm paraffin ileal sections were rehydrated. Antigen retrieval was performed by boiling in a pH 6 citrate buffer with 0.5% Tween-20, and then incubated with blocking solution (5% goat serum in PBS) for one hour at room temperature. For tight junction protein staining, tissue sections were incubated with Alexa Fluor 647-conjugated ZO-1 mouse monoclonal antibody (1:100, Thermo Fisher, Waltham, MA, USA, #MA3-39100-A647) and Alexa Fluor 594-conjugated occludin mouse monoclonal antibody (1:100, Thermo Fisher, #331594) overnight at 4 °C. For staining, the activated NF-κB, sections were incubated with primary antibody solution (rabbit polyclonal to phospho S529b NF-κB p65, 1:100, Abcam, Cambridge, UK, ab97726) overnight at 4 °C. After washing with PBS three times, the sections were incubated with Alexa Fluor™ 594-conjugated secondary antibody (1:1000, Thermo Fisher, # A11012) for one hour at room temperature. All sections were washed three times with PBS and then counterstained and mounted with ProLong™ Gold Antifade Mountant with DAPI (Thermo Fisher, # P36935). Images were captured with a Stellaris confocal microscope (Leica Microsystems, Inc., Buffalo Grove, IL, USA). ImageJ (U. S. National Institutes of Health, Bethesda, MD, USA, http://imagej.nih.gov/ij/, 1997–2012, accessed on 28 February 2023) was used for imaging processing and analysis.

### 2.6. Cytokine Assay

Mice at two weeks of age were euthanized under isoflurane, followed by cardiac puncture for blood collection. Blood samples were spun down at 2000× *g* for 10 min. Supernatants were collected as sera and stored at − 80 °C for analysis. Multiplex analysis was performed according to the manufacturer’s instructions using a kit for a panel of mouse cytokines/chemokines based on the Luminex^®^ xMAP^®^ technology with magnetic beads (EMD Millipore Corporation, Billerica, MA, USA). Experiments were performed in triplicate.

### 2.7. Behavioral Studies

Tests were conducted at four weeks of age as previously described [33] and were recorded and analyzed using ANY-maze software (V7.1) (Stoelting Co., Wood Dale, IL, USA).

Open Field Test: Animals were placed individually in the center of an open field box, and their spontaneous motor activity was recorded in terms of the following parameters: mean speed, total time immobile, and time spent travelling in the center and border zones. 

Elevated-plus maze: The elevated plus maze consists of four arms and a central area elevated 50 cm above the floor. Two arms are open and two are closed with walls. Mice were individually placed in the center facing an open arm and allowed to explore for five mins. The following parameters were recorded: time spent in the closed and open arms.

Contextual and cued fear conditioning test: The contextual and cued fear conditioning tests the ability of mice to learn and remember an association between environmental cues and aversive experiences. In this test, mice were placed into a conditioning chamber and were given parings of a CS (an auditory cue) and an aversive US (an electric foot shock). During the conditioning stage at day 1, mice were allowed to freely explore the chamber for 120 s. Thereafter, a white 55 dB noise auditory cue was presented as a CS for 30 s, and a 0.8 mA foot shock was given to the mice as an US continuously during the last two seconds of the sound. The presentation of CS-US was repeated three times per session (120, 240, and 360 s after the beginning of the conditioning). Following the final foot shock, the mice were left undisturbed in the chambers for 90 s. After the conditioning session had been completed, the mice were returned to the same conditioning chamber 24 h later and were allowed to freely explore the chamber for 300 s without CS and US presentations. 

The cued test was conducted on the same day two hours after the context test in a novel context that was unrelated to the conditioning chamber. Mice were placed into the testing chamber for three minutes. At the end of the first three minutes, the CS auditory cue that had been presented at the time of conditioning was given to mice for three minutes. Fear memory was assessed based on freezing behavior to the conditioned cued or the contexts to which mice were previously exposed. The outcome variables were freezing time in the context test and during the last 300 s of the cued test.

### 2.8. MRI

#### 2.8.1. In Vivo Macromolecule Proton Fraction (MPF) Imaging

Imaging of four-week-old mice was performed on a 14.1 T Bruker Avance imaging spectrometer (Bruker, Billerica, MA, USA) using a 20-mm resonator as previously described [33]. MPF maps (Figure 1) were obtained from three images (Magnetization transfer (MT)-, Proton density (PD)-, and T1-weighted) using the single-point method with the synthetic reference image [46]. All images were acquired in the axial plane with whole-brain coverage and a resolution of 0.125 × 0.125 × 0.25 mm^3^. In all 3D imaging experiments, linear phase-encoding order with 100 dummy scans, slab-selective excitation, and fractional (75%) k-space acquisition in the slab selection direction were used. To correct for field heterogeneities, two 3D maps were acquired using the dual-TE (TR/TE1/TE2 = 20/2.9/5.8 ms, α = 8°) and actual flip-angle imaging (AFI) (TR1/TR2/TE = 13/65/4 ms, α = 60° methods, respectively [47]. All reconstruction procedures were performed using custom-written C-language software athttps://www.macromolecularmri.org/, accessed on 28 February 2023 (V2.0).

#### 2.8.2. Ex Vivo MRI

Mice were fixed with transcardial perfusion of saline followed by 4% paraformaldehyde (Sigma Aldrich, St. Louis, MO, USA). Brains remaining in skulls were immersed in non-aqueous media (fomblin Y, Sigma Aldrich, St. Louis, MO, USA). Ex vivo MR imaging was performed on a 14.1 T Avance imaging spectrometer (Bruker) using a 30-mm resonator. Diffusion tensor imaging (DTI) experiments consisted of 30 non-collinear direction diffusion weighted spin echo images. The imaging parameters were TR/TE/NEX 7500/15.1/1, 40 axial slices 0.25 mm thick with no gap covering the whole cerebrum. In-plane resolution was 125 × 125 µm, and brain volumes were interpolated to isotropic 125 µm^3^. Diffusion tensor maps were calculated using multivariate linear fitting of signal attenuation from the acquired diffusion weighted images [48]. Fractional anisotropy (FA) maps were calculated [49] using in-house software written on Matlab (V1.0) (MathWorks, Natick, MA, USA).

#### 2.8.3. MRI Data Processing

Magnetization transfer images of mouse brains, obtained as a component of the in vivo MPF experiment and possessing excellent white/gray matter contrast, were used for automatic structural parcellation using the multi-atlas label fusion method, as detailed in [50]. Briefly, individual mouse head images were processed for brain extraction, intensity non-uniformity correction, affine registration to atlas images, and label fusion. The publicly available MRMNeAt atlas database was used, containing 10 individually labeled C57BL/6J in vivo mouse brains. The values of MPF were compared between MTERRM (*n* = 5) and MNEC (*n* = 5) groups.

For the analysis of ex vivo DTI data, a cross-subject voxel-wise Tract Based Spatial Statistics analysis (TBSS) [51,52] was utilized, as implemented in the FSL software (V6.0) (http://www.fmrib.ox.ac.uk/fsl/, accessed on 8 February 2023). All individual FA volumes were registered to a template, and the mean FA-map was calculated and thinned to represent the mean FA skeleton. For each subject, voxel data were projected from FA maps to the nearest voxels on the mean FA skeleton. The values of voxels on the common skeleton were analyzed with voxel-wise cross-subject statistical analysis utilizing a general linear model [51,52]. In the case of two groups, as in this study, testing the contrast between the group predictors is equivalent to an unpaired *t*-test of the mean difference between the groups. As a result of the procedure, statistical parametric maps were created containing *p*-values for the voxel-wise two-sample unpaired *t*-tests. The results were corrected for multiple comparisons by controlling the family-wise error rate.

### 2.9. RNA Isolation and Real-Time PCR

Using the RNeasy^®^ Plus Mini Kit (QIAGEN GmbH, Hilden, Germany), total RNA from snap frozen brains was isolated. An amount of 500 ng of isolated total RNA was used to synthesize cDNA using an RT^2^ First Strand Kit from QIAGEN. TaqMan^®^ single tube Assays (Thermo Scientific Inc., Waltham, MA, USA) were used for genes of interest and the housekeeping gene *Gapdh*. Gene expression of interest was normalized to the housekeeping gene.

### 2.10. Metabonomic Analysis

#### 2.10.1. Data Collection

Brain samples were submitted to the Microbiome Metagenomics Facility of the DFI for metabolite extraction. Metabolites were extracted with the addition of extraction solvent (100% methanol spiked with heavy labeled bile acid and amino acid internal standards and stored at −80 °C) to brain samples at a ratio of 100 mg of material per 1000 µL of extraction solvent in beadruptor tubes. Samples were homogenized at 4 °C on a Bead Mill 24 Homogenizer (Fisher; 15-340-163) and then incubated at −20 °C overnight. Samples were then centrifuged at −10 °C, 20,000× *g*, for 15 min, and 75 µL of the supernatant was transferred to a microcentrifuge tube. Samples were dried down completely using a Genevac EZ-2 Elite. Samples were resuspended in 150 µL of 50:50 water:methanol and added to an Eppendorf thermomixer^®^ at 4 °C, 1000 rpm for 15 min, to resuspend analytes. Samples were then centrifuged at 4 °C, 20,000× *g*, for 15 min to remove insoluble debris. AN amount of 100 µL of supernatant was transferred to a prelabeled MS vial with insert.

#### 2.10.2. Untargeted Data Collection

Samples were analyzed on a Thermo Fisher liquid chromatography system coupled to an Orbitrap IQ-X mass spectrometer. The mobile phase A was water with 0.1% Formic Acid and mobile phase B was 95% Acetonitrile with 0.1% Formic Acid. The electrospray ionization conditions were set with the spray voltage at 3.5 kV, vaporizer temp at 400 °C, and detection window set to 133–2000 m/z. Precursor selection for MS^2^ scans was set to 133–2000 m/z with dynamic exclusion after 3 times within 15 s for a duration of 10 s. The isolation window was 0.7 m/z, with no offset and a fixed normalized collision energy of 35%.

#### 2.10.3. Bile Acid Analysis

Bile Acids were analyzed using LCMS. Samples were analyzed on an Agilent 1290 infinity II liquid chromatography system coupled to an Agilent 6546 QTOF mass spectrometer, equipped with an Agilent Jet Stream Electrospray Ionization source. An amount of 5 µL of sample was injected onto an Xbridge© BEH C18 Column (3.5 µm, 2.1 × 100 mm; Waters Corporation, 186003022) fitted with an XBridge© BEH C18 guard (Waters Corporation, 186007766) at 45 °C. The mobile phase A was water with 0.1% Formic Acid and mobile phase B was Acetone with 0.1% Formic Acid. The electrospray ionization conditions were set with the capillary voltage at 3.5 kV, nozzle voltage at 2 kV, and detection window set to 100–1700 m/z with continuous infusion of a reference mass (Agilent ESI TOF Biopolymer Analysis Reference Mix) for mass calibration. A ten-point calibration curve was prepared with 62.5 µg/mL each in water of Cholic Acid, Taurocholic Acid, Glycocholic Acid, Deoxycholic Acid, 3-Epideoxycholic Acid, 3-oxolithocholic Acid, Alloisolithocholic Acid, Lithocholic Acid, Chenodeoxycholic Acid, Glycochenodeoxycholic Acid, Taurochenodeoxycholic Acid, Ursocholic Acid, Ursodeoxycholic Acid, alpha-Muricholic Acid, beta-Muricholic Acid, and Allocholic Acid, with nine subsequent 3× serial dilutions. Data analysis was performed using MassHunter Profinder Analysis software (version B.10, Agilent Technologies, Santa Clara, CA, USA) and confirmed by comparison with authentic standards. Normalized peak areas were calculated by dividing raw peak areas of targeted analytes by averaged raw peak areas of identical or most structurally similar internal standards.

#### 2.10.4. Data Processing 

Raw data files were converted into open-source file format and processed using MZmine2 and the Feature-Based Molecular Networking function in the Global Natural Products Social Molecular Networking (GNPS) environment to identify features and match data to publicly available library spectra.

#### 2.10.5. MZmine

MZmine 2.53 [53] was used to create feature lists with abundances in each sample from the raw data. The settings used were based on manual inspection of the raw data for values that represented signals above the inherent noise level, typical peak shapes, and mass and retention time (RT) tolerances. First, a mass detection was used with a noise cutoff filter for both MS1 and MS2 scans to create a mass list for each data file. The Centroid mass detector was set to a level of 6.0 × 10^3^ for the MS2 level and 1.0 × 10^4^ for the MS1 level. The ADAP chromatogram builder algorithm was used to create extracted ion chromatograms at the MS1 level, with a minimum group size of 3 scans, group intensity threshold of 1.0 × 10^4^, minimum highest intensity of 3.0 × 10^4^, and m/z tolerance of 0.015 Da or 5.0 ppm. Chromatogram deconvolution was performed using the Wavelets (ADAP) algorithm. The isotopic peak grouper module was used to group features that were isotopes. A master feature list was created using the Join aligner module. The resulting feature list was filtered to remove duplicate features, and then the gap-filling algorithm was used to fill in any missing values for peaks that were not detected with the previous algorithms. The peak-finder gap filling algorithm was used with an intensity tolerance of 10%, m/z tolerance 0.02 Da or 5.0 ppm, and RT tolerance 0.1 min. Peaks were filtered to remove any with a peak area less than 3.0 × 10^4^. The resulting list was exported for GNPS analysis. 

#### 2.10.6. The Global Natural Product Social Molecular Networking (GNPS)

A molecular network was created with the Feature-Based Molecular Networking (FBM N) workflow [54] on GNPS (https://gnps.ucsd.edu) [55], accessed on 22 February 2023. The results from MZmine2 were exported to GNPS for FBMN analysis. The data was filtered by removing all MS/MS fragment ions within +/−17 Da of the precursor m/z. MS/MS spectra were window filtered by choosing only the top 6 fragment ions in the +/−50 Da window throughout the spectrum. A molecular network was then created, where edges were filtered to have a cosine score above 0.7 and more than 5 matched peaks. Further, the edges between two nodes were kept in the network if and only if each of the nodes appeared in each other’s respective top 10 most similar nodes. Finally, the maximum size of a molecular family was set to 100, and the lowest scoring edges were removed from molecular families until the molecular family size was below this threshold. The analogue search mode was used by searching against MS/MS spectra with a maximum difference of 100.0 in the precursor ion value. The library spectra were filtered in the same manner as the input data. All matches kept between network spectra and library spectra were required to have a score above 0.7 and at least 5 matched peaks. The job can be accessed at https://gnps.ucsd.edu/ProteoSAFe/status.jsp?task=2fe38a45d2d345c4b4c137fcd16d8980, accessed on 22 February 2023.

#### 2.10.7. Feature List Filtering

The feature list from GNPS was exported with putative IDs associated and filtered in Excel using the peak areas found in blanks and quality control injections to remove low quality features from the statistical analysis. Features within a sample were normalized by dividing each peak area by the average peak area of the internal standards in that sample. The internal standards used for normalization spanned the retention time gradient and included D4-cholic acid, D4-taurocholic acid, D4-glycocholic acid, D4-deoxycholic acid, D4-chenodeoxycholic acid, 13C9-tyrosine, 13C9-pheylalanine, and 13C11-tryptophan (Cambridge Isotope Laboratories, Tewksbury, MA, USA). A pooled QC of all samples was injected throughout the run, and 3 injections of the QC were used to calculate the percent coefficient of variation (%CV) of features. Any features with normalized peak area %CV of greater than 20% across the 3 pooled QC injections were eliminated. The total feature list contained 1492 features for all unknown features. This list was further analyzed to consider only features with a putative ID assigned, and from that list any features that GNPS matched to known mass spectrometry contaminants and polyether polymers were removed, resulting in 231 features for all putatively classified features. 

#### 2.10.8. MetaboAnalyst

MetaboAnalyst [56] was used to statistically analyze and visualize both the full unknown feature list and the putatively classified features. For each subset, any features that had a single or constant value across all samples were removed. MetaboAnalyst filtering and normalization steps were not applied because these steps were performed previously as described. Figures were exported from MetaboAnalyst.

### 2.11. Statistical Methods

R statistical software (v4.0.3) was used to perform microbial analysis. To correct for multiple testing, the Benjamini–Hochberg method was utilized to evaluate which tests passed a false-positive threshold of <0.05%. For other statistical analysis, GraphPad Prism 9.5 (La Jolla, CA, USA) was used to test the difference. Student *t*-test or Mann–Whitney U test were used for two groups, with *p* values < 0.05 considered to be significant.

## 3. Results

### 3.1. Dysbiotic Nature of Microbial Communities in Offspring Colonized with Fecal Samples from Patients Who Went on to Develop NEC

Fecal samples were collected from pups colonized with human fecal samples from term infants (MTERM) and preterm infants who went on to develop NEC (MNEC) at two and four weeks of age. Samples from MTERM (*n* = 8) and MNEC (*n* = 6) pups at two weeks of age and MTERM (*n* = 13) and MNEC (*n* = 7) pups at four weeks of age were subjected to 16S rRNA sequencing. There was significant difference in species richness and evenness (α-diversity) measured by Inverse Simpson (InvSimpson) (*p* < 0.05) and Shannon (*p* < 0.001) diversities between MTERM and MNEC fecal samples at two weeks of age (Figure 2a). The measured α-diversity indices at four weeks of age between the two groups were not significantly different. We then evaluated microbiome sample clustering using the Bray–Curtis dissimilarity metric based on all operational tax taxonomic units (OTUs). PERMANOVA analysis, as reflected in the principal coordinate analysis plot (PCoA), revealed that there was an overall significant separation (β-diversity) in the fecal microbial communities among the four groups by PERMANOVA (Figure 2b, *p* = 0.001). Pairwise, *post hoc* tests revealed that β-diversities were statistically significant between MTERM and MNEC fecal samples at two and four weeks old (both at adjusted *p* = 0.002), suggesting an overall compositional difference among the two treatment groups at both ages.

We further evaluated specific differential abundances of OTUs on the genus level between the two treatment groups by MaAsLin2 [57] in R package, with FDR adjustment value of 0.05. The abundance of *Enterococcus* from phylum *Bacillota* was significantly decreased, and *Bacteroides* from phylum *Bacteroidota* and *Escherichia. Shigella* from phylum *Pseudomonadota* was significantly enriched in the MNEC group compared to the MTERM group at two weeks of age (Figure 3a–c). MNEC mice had a significantly greater abundance of *Acetitomaculum*, *Anaerostipes*, *Murimonas*, *Enterococcus*, *Ligilactobacillus*, *Faecalibaculum* from phylum *Bacillota*; *Bacteroides*, *Duncaniella*, *Muribaculum* from phylum *Bacteroidota*; *Parasutterella* from phylum *Pseudomonadota*; and *Akkermansia* from phylum *Verrucomicrobiota* than MTERM mice at four weeks of age (Figure 3d–i,k–m,o,p). The abundance of *Alistipes* from phylum *Bacteroidota* and *Escherichia.Shigella* from phylum *Pseudomonadota* was significantly reduced in MNEC mice at four weeks of age (Figure 3j and Figure 3n, respectively). These data demonstrated that MNEC microbial communities were distinctly different from those of MTERM in α-diversity at two weeks of age, overall bacterial community composition, and specific taxonomic abundance levels at both two and four weeks of age.

### 3.2. Mice Colonized with Fecal Samples from Preterm Infants Who Developed NEC Displayed Decreased Tight Junction Development and Increased Inflammation in the Small Intestine

Because a dysfunctional intestinal barrier characterized by irregular tight junctions at the intercellular spaces of intestinal epithelial cells is a strong indicator of gut immaturity and is associated with microbial translocation from the gut as well as inflammatory responses [21,58,59], we investigated the effect of the two microbial communities used to colonize the host on the development of tight junctions in the distal small intestine (ilea), the area of the intestine classically affected by NEC. Using immunohistochemical analysis, we detected, when compared to MTERM mice (Figure 4a), a significantly lower surface expression of occludin and ZO-1 in MNEC mice (Figure 4b) (quantification in Figure 4c,d, both groups *n* = 3, *p* < 0.05, and *p* < 0.001, respectively) at two weeks of age.

Using NF-κB activation (defined as nuclear translocation of the phosphorylation of the p65 subunit) as a marker for inflammation, we demonstrated that in the crypt of the ileum, mice colonized with MTERM (Figure 5a) had significantly decreased nuclear phospho-p65 subunit of NF-κB compared to MNEC mice (Figure 5b) (with quantification in Figure 5c, both groups *n* = 4, *p* < 0.01). There was no difference in the nuclear presence of the phosphorylated p65 subunit between MTERM and MNEC in the villus of the ileum (Figure 5d, *p* > 0.05). These data demonstrate that microbial communities from NEC patients can result in mucosal barrier development deficiency and intestinal inflammation in the small intestine.

### 3.3. Systemic Inflammation Was Not Different in Offspring Colonized with Fecal Samples from Preterm Infants Who Developed NEC When Compared to Those from Term Infant

Because systemic inflammation is strongly elevated in NEC patients compared to the healthy controls [60], we evaluated the difference in systemic inflammation between MTERM and MNEC mice at two weeks of age. In two-week-old mice, the serum levels of TNF, KC, and IL-1α evaluated by multiplex ELISA were similar between MTERM and MNEC mice (Figure 6a–c, all *p* > 0.05, *n* = 7–8). There was also no difference in the levels of IL-6, IL-1β, MIPα, and IL-10 between the two groups. 

### 3.4. Mice Colonized with Fecal Samples from Preterm Infants Who Developed NEC Displayed Behavioral Deficits

#### 3.4.1. MNEC Mice Exhibit Decreased Locomotion and Decreased Anxiety

We evaluated the effects of microbiota on general locomotor activities and anxiety-like behaviors using the open field test (OFT) and the elevated plus maze (EPM) test. MTERM mice were the more mobile (immobile time = 61.1 ± 18.6 s(s), *n* = 12) than MNEC mice (immobile time = 115.6 ± 15.7 s, *n* = 11) (Figure 7a, *p* < 0.01) in the OPT. MTERM mice had significantly higher mean speed (0.037 ± 0.002 m/s) than MNEC mice (0.028 ± 0.002 m/s) (Figure 7b, *p* < 0.01). MTERM mice spent significantly less time (1039 ± 18.4 s) than MNEC mice (1125 ± 11.8 s) in the border zone of the open field (Figure 7c, *p* < 0.001). MTERM mice also spent significantly more time (123.2 ± 15.3 s) than MNEC mice (74.9 ± 11.8 s) in the center zone (Figure 7d, *p* < 0.05). Furthermore, in the EPM test there was no difference in time spent in the closed arms between four-week-old MTERM and MNEC mice (Figure 7e). MTERM spent significantly less time in the open arms (22.6 ± 4.3 s, *n* = 7) than MNEC mice (54.2 ± 10.2 s, *n* = 9) in the EPM test (Figure 7f, *p* < 0.05). These data demonstrate that mice colonized with microbial communities from infants who went on to develop NEC developed locomotion deficits and decreased anxiety when compared to mice colonized with communities from term infants.

#### 3.4.2. MNEC Mice Exhibit Impaired Contextual Memory

To investigate the effects of microbiota on associative learning and memory, MTERM and MNEC mice were subjected to the contextual/cued fear conditioning test. MTERM mice had significantly longer freezing times (184.2 ± 19.2 s) than MNEC mice (55.4 ± 7.4 s) at four weeks in the contextual fear conditioning (same environment as the conditioning session) (Figure 8a, *p* < 0.0001). No difference in freezing time in the second half of the cued fear conditioning test (changed environment and acoustic CS presentation) was found at four weeks of age (Figure 8b). These data demonstrated that mice colonized with microbial communities from preterm infants who went on to develop NEC acquired contextual memory impairment.

### 3.5. Mice Colonized with Fecal Samples from Preterm Infants Who Developed NEC Displayed Deficits in Brain Maturation

Because we have previously demonstrated that myelination and white matter organization can be affected by gut microbiota and that both are strongly associated with postnatal neurologic functional maturity [33,61], we evaluated the effects of the two microbial communities on myelination and white matter organization using macromolecular proton fraction (MPF) mapping (see Figure 1 in Method) and diffusion tensor imaging (DTI). MNEC mice had significantly decreased MPF in corpus callosum, internal capsule, neocortex, and cerebellum when compared to MTERM mice at four weeks of age (Figure 9a–d, respectively, both *n* = 5, with significant *p* values displayed in the graphs), indicating a significant hypomyelination in these white and grey matter tracts. MNEC mice displayed significantly lower fractional anisotropy (FA) values measured by DTI in the corpus callosum (cc), internal capsule (ic), and medial lemniscus (ml) (Figure 10) when compared to MTERM mice at four weeks of age, further demonstrating delayed maturation and poor organization in white matter tracts in the MNEC mice.

### 3.6. Mice Colonized with Fecal Samples from Preterm Infants Who Developed NEC Displayed Cellular Deficits in Brain Development

To examine whether microbial colonization impacts early brain development, we measured the transcriptional levels of microtubule-associated protein 2 (MAP2), a marker of post-mitotic mature neurons at the early developmental stage of the brain, by RT-PCR. The brain mRNA expression levels of *Map2* were significantly lower in MNEC mice when compared to MTERM mice (Figure 11a, both *n* = 6, *p* < 0.01), demonstrating that there is a microbiota-dependent delay of neuronal maturation. Oligodendrocyte progenitor cell marker Olig2 (Figure 11b) and astrocyte marker (S100B) (Figure 11c) were not different between the two groups. There was a significant difference of pericyte marker (PDGFRβ) expression between the two groups (Figure 11d, *p* < 0.05). These data suggest that the microbial communities from infants who went on to develop NEC negatively impacted neuronal and pericyte development.

### 3.7. Microbial Communities from Preterm Infants Who Developed NEC Significantly Shifted the Metabolite Features in the Brain

To determine whether the two sources of the bacterial communities in our study induce distinct metabolic response in the brain, we subjected brain samples to both non-targeted and targeted metabolite profiling. The non-targeted metabolomics experiment resulted in 1492 features. To test if overall brain metabolic profiles were affected by the two different microbial groups, a principal component analysis (PCA) was used to perform unsupervised clustering based on feature abundances. At two weeks old, the metabolic profiles in the brain samples (Figure 12a, both *n* = 6) were not separated between the two treatment groups. Features were then computationally assigned putative molecular IDs using the GNPS online platform, and only features that could be assigned an analogous match to fragmentation spectra found in the library were included in the analysis. Feature list filtering, as described in the Methods, resulted in 231 high-quality features with putative IDs. The distinct compositional difference by treatments at two weeks of age is demonstrated in the volcano plots of significantly differential abundance of the features based on *t*-test shown with FDR (cutoff at 0.05)-adjusted *p* value (Figure 12b) and without FDR adjustment (Figure 12c). Two features with the most significant difference were putatively identified as O-arachidonoylcarnitine and 1,2-Dihexadecanoyl-sn-glycerol-phosphocholine. We further revealed that three analogs of carnitine, carnitine, acetylcarnitine, and arachidonoylcarnitine, but not myristoylcarnitine, lauroylcarnitine, or palmitoylcarnitine, were significantly higher in the brains of the MNEC mice than those in the MTERM mice (Figure 12d). There were also three phosphocholine analogs (isomers 1 and 2 of diPamitoylphosphophatidylcholine and 1-(eicosatetraenoyl)-2-octadecanoyl-glycero-3-phosphocholine) in the brain metabolite pool that were significantly lower in MNEC mice (Figure 12e). Using a 40-standard panel of bile acid analogs, we identified that the normalized levels of taurocholic acid (Figure 12f) and tauro-alpha or tauro-beta muricholic acid (Figure 12g) were significantly lower in the brain samples of MNEC mice.

## 4. Discussion

NEC and its associated NDI remains a substantial challenge for healthcare professionals and patients. Gut microbiota and its interplays with an immature gut barrier with a disrupted tight junction that constructs the paracellular space between the enterocytes have been proposed as the main contributing factors to the development of NEC and NDIs in NEC patients [62,63,64]. In this study, we isolated the effects of microbial communities from human preterm infants who developed NEC on the brain development and neurological outcomes using a “humanized” gnotobiotic model in C57/BL6J mice. Mice colonized with microbial communities from infants who went on to develop NEC had deficits in mucosal barrier development, brain maturation, locomotion, anxiolytic behaviors, and contextual memory when compared to mice colonized with microbial communities from healthy term infants, demonstrating that gut microbiota from patients before the onset of NEC drives the neurodevelopment outcomes.

Aberrant bacterial colonization preceding NEC has been documented in clinical studies [23,24,25]. In our mice colonized with fecal samples from preterm infants who went on to develop NEC, sequencing analysis showed that when compared to MTERM mice, MNEC mice at two weeks of age had reduced microbial α-diversity and altered composition of gut microbiota signified by decreased *Enterococcus* from phylum *Bacillota* (previously *Firmicutes*) and enriched *Escherichia. Shigella* from phylum *Pseudomonadota* (previously *Proteobacteria*), which was consistent with the clinical reports. At four weeks of age, there was an expansion of genera from phylum *Bacillota* in MNEC mice compared to MTERM mice, reflecting a prolonged dominance of phylum *Bacillota,* which is characteristic of a delayed maturation of preterm infant microbiota when compared to term infants [22]. Overall, mice colonized with microbial communities from preterm infants before the onset of NEC displayed distinct dysbiotic patterns when compared to the mice colonized with microbial communities from term infants.

Enterocytes in the gastrointestinal tract with a well-organized physical barrier represent the first defense line against translocation of gut luminal bacterial and other toxin molecules [65]. Intestinal barrier failure, either as a prerequisite or a result of disease development, is a phenotypical feature of NEC [66]. The proper function of the paracellular barrier is maintained by an orchestrated complex with several tight junction proteins, including occludin, zonula occludens (ZOs), tricellulin, cingulin, and junctional adhesion molecules [67]. In light of the studies reporting that intestinal expression of the tight junction proteins occludin and ZO-1 were significantly reduced in NEC patient specimens [66] and in a rat model of NEC [68], we examined the expression of occludin and ZO-1 under the influence of microbial communities from NEC patients. We discovered that microbial communities from patients who went on to develop NEC significantly down-regulated the villous expression of occludin and ZO-1 in the ileal tissues. Furthermore, we observed increased crypt inflammation marked by increased NF-κb activation in the mice colonized with microbial communities from NEC patients. Specialized secretory epithelial cell types such as Paneth cells located in the base of intestinal crypts limit bacterial adhesion and infiltration by secreting antibacterial peptides [69]. However, inflamed Paneth cells can also mount immune responses, particularly releasing tumor necrosis factor, through pathogen-associated molecular pattern molecules or bacterial invasion to induce microvascular injury in the small intestine, which eventually lead to necrosis of the intestinal villi [70]. The fact that the reduced expression of occludin and ZO-1 in the villus and increased crypt inflammation we observed were not under the presence of diseased states as previously seen in NEC patients and animal NEC tissues, reveals that gut microbiota can directly affect the development of mucosa barrier development and small intestine homeostasis.

Infants who develop NEC suffer long term NDIs [71]. In an animal model of NEC when pathogenic bacteria were introduced to mice, mice developed spatial working and novel recognition memory impairments [72]. Our previous studies demonstrated that commensal bacteria are required for normal brain development and behaviors [33] and prematurity of preterm infant microbiota negatively impacts neurological outcomes [42]. In this study, we demonstrated that motor abilities reflected by mobility and speed of movement, anxiolytic behaviors reflected in both the open field and elevated plus maze tests, and contextual memory in fear conditioning test were significantly impaired in the mice colonized with microbial communities from infants who went on to develop NEC. These results are first in agreement with the growing evidence illustrated by our previous studies and others that initial bacterial colonization of gut microbiota plays a significant role in determining the outcomes of neurological development, and preterm infants might be at specific risk for adverse outcomes due to the prematurity of gut microbiota [42]. Secondly, the initial colonization of microbial communities from patients who went on to develop NEC in GF mice resulted in deficits in NDIs, such as cognitive impairment and psychomotor impairment resembling those in the NEC population, substantiating a causative effect of microbial communities on driving NDIs in NEC infants.

Myelination is a hallmark of brain maturation and is significantly associated with cognitive function development in developing brains. Evaluation of the maturation of the brain can be achieved by examining the cerebral cortex and white matter myelination [73,74,75], and myelination of the brain in these areas strongly correlates with postnatal neurologic functional maturity [61]. NEC-associated brain injury is associated with loss of myelin in both animal model and in human tissue of NEC patients [72]. We have previously also shown that mice lack of commensal bacteria had profound hypomyelination in major white tracts, including corpus callosum, anterior commissure, and internal capsule, as well as in gray matter structures, including the neocortex, hippocampus, hypothalamus, and brainstem/midbrain [33]. We further established that myelination closely corresponds to MPF values across the developmental span in white matter and the cortex, with Pearson correlations between MPF and myelination across the developmental span being 0.90 for white matter and 0.79 for the cerebral cortex, respectively [76]. Using MPF as a measurement for myelination and brain maturation, in this study we demonstrated that mice colonized with microbial communities from preterm infants who went on to develop NEC had significant hypomyelination in the brain regions of the corpus callosum, internal capsule, neocortex, and cerebellum. Neocortex affords higher brain cognitive functions, including learning and memory, planning complex cognitive behavior, and moderating social behavior [77,78]. The cerebellum is the motor learning site of the CNS [79] and defects in myelination lead to motor deficits and motor learning deficits [80]. The DTI measurement of FA values confirmed the delayed maturation and poor organization in the white matter tracts such as the corpus callosum and internal capsule in mice colonized with microbial communities from patients who went on to develop NEC. The FA values of corpus callosum and internal capsule have been shown to be positively associated with cognitive outcomes and motor skills [81,82,83,84,85]. In human studies, FA values of white matter tracts, including the corpus callosum, correlate positively with fine motor and cognitive Bayley scores in children born prematurely [86]. Therefore, we speculate that the microbiota-dependent differences we observed in the myelination and maturation of cortex, cerebellum, corpus callosum, and internal capsule might explain the differences in motor activity in the open field test and contextual memory in the fear conditioning test in our study.

At the molecular level, MNEC mice also displayed insufficient development of mature neurons and pericytes, evidenced by the lower transcriptional levels of Map2 and PDGFRβ. Of the suggested potential influences of gut microbiota on CNS, systemic inflammation, immune surveillance, and production of metabolites/neuromodulators/neurotransmitters represent the most investigated mechanisms [87]. Gut microbiota-derived systemic inflammation through activation of immune systems in NEC patients has been proposed to be one of the mechanisms for the development of NDI [60,71,88,89]. In this study we did not detect significant difference in serum proinflammatory mediators between the MTERM and MNEC groups. The reasons could be several fold: (1) we measured the inflammatory markers at two weeks of age, and the systemic inflammation could have occurred at birth or earlier in life; (2) most of the systemic inflammation reported occurred at the time point at which infants developed NEC. Our model is not an NEC model per se and was designed to study the effects of the microbial communities before the onset of NEC on neurodevelopment; (3) both groups of mice were breast fed by the dams, which would not happen in most of the preterm infants or NEC patients. In human studies, breastfeeding or an exclusive human milk diet is associated with reduced risk for neonatal infection and NEC [90,91,92]. It is important to point out that many mothers with preterm delivery have difficulty providing adequate amounts of breast milk or breast milk with adequate nutritional contents [90]. In fact, nutritional content in preterm infants’ diet is a key regulator of neurodevelopment outcome independent of neonatal infection [90], suggesting that modifiable factors such as diet or microbiota as we identified in this study might collectively contribute to the neurodevelopment outcomes of preterm infants.

We therefore sought to look for the other potential features leading to the altered CNS development and functions we observed in this study. Gut microbiota-induced metabolic responses have been shown to be involved in gut microbiota and brain communication [93]. In our study we identified two features with putative IDs that are phosphocholine analogs and carnitine analogs that were different between MTERM and MNEC brain samples. Phosphatidylcholines are both the major component of cell membranes and precursor of sphingomyelin, which is responsible for insulating the axons of neurons to ensure transmission of neuronal circuit signals [94,95]. In addition, the amount of phosphatidylcholine is positively correlated with neuronal cell differentiation, which is dependent on the membrane biosynthesis to support neuritogenesis and neurite outgrowth [96]. Furthermore, hydrolysis of phosphatidylcholine also delivers the precursor choline for acetylcholine, which is the neurotransmitter responsible for the cholinergic neural circuit associated with sense, recognition, motion, learning, and memory [97,98]. In fact, a deficiency of choline during the perinatal period leads to compromised cognitive function [99]. Our results demonstrated that colonization of microbial communities from preterm infants before the onset of NEC in mice resulted in a significantly reduced abundance of phosphocholine and phosphatidylcholine, which thus suggested that the observed deficits in behaviors might be due to the decreased availability of phosphocholine/phosphatidylcholine related to choline metabolism.

We further observed that levels of several carnitine analogs including carnitine, acetyl-carnitine, and arachidonoylcarnitine (acylcarnitine) were increased in the brains of mice colonized with microbial communities from preterm infants before the onset of NEC. Carnitine and acetylcarnitine treatments have been shown to reduce memory loss in aging rodents [100,101]; however, the efficacy of L-carnitine supplementation for the enhancement of cognitive function in people without cognitive impairment or to support growth in preterm infants was not substantiated [102,103]. In fact, preterm infants with postnatal growth failure and malnourished children displayed a metabolic profile of elevated acylcarnitine, other carnitine analogs, and β-oxidation product ketone body [104,105]. Furthermore, mice colonized with fecal samples from malnourished children also developed a metabolic profile marked by increased levels of several acylcarnitine analogs [106]. Carnitine analogs such as arachidonoylcarnitine function as carriers to transport long chain fatty acids into mitochondria for β-oxidation to generate ATP and shuttle the medium and short chain fatty acids out of mitochondria [107]. However, in normal physiological conditions, the brain mainly relies on glycolysis for its energy requirement to support brain growth and functions [108]. Impaired cerebral glucose uptake due to glucose transporter type 1 deficiency has been reported to result in clinical symptoms such as seizures, movement disorders, and cognitive impairments [109]. We speculate that the elevated levels of several carnitine analogs in MNEC brains in this study might reflect the metabolic adaptation of energetic deficit elicited by dysregulated glucose availability and/or metabolism in the brain.

Bile acids and metabolic products of bile acids have also been suggested to participate in the communication between the gut microbiota and the brain [110]. We demonstrated that the modulation of offspring microbiota during lactation through maternal supplementation of probiotics induced taurocholate, taurocholic acid, and taurohyocholic acid crossing the blood–brain barrier [43]. In this study we have identified that taurine-conjugated choline and tauro-α/β muricholic acid levels were significantly higher in MTERM mice than those in the MNEC mice. The specific roles of these bile acids in the CNS are yet to be explored. Suggested impacts rely on the facts that the brain expresses membrane Takeda G protein-coupled receptor 5 (TGR5) and nuclear farnesoid X receptor (FXR) [111]. Rodent taurine-conjugated bile acids have anti-apoptosis effects on neurons through its brain TGR5 receptor [112]. Tauro-β-muricholic acid (MCA), a potent FXR antagonist, regulates energy metabolism in obesity, insulin resistance, and nonalcoholic fatty liver disease [113]. Therefore, we propose that the deleterious effects of microbial communities on MNEC mice in the observed behaviors might be due to NEC microbial communities-induced dysregulation of bile acid-related metabolism.

## 5. Conclusions

In conclusion, we have demonstrated that microbiota from human preterm infants who went on to develop NEC drives deficits in brain development and neurological outcomes in offspring in a humanized mouse model. Preterm infants with NEC might be at specific risk for delayed brain maturation, locomotion deficits, anxiety, and impaired contextual memory. The changed metabolic profile identified in this study provides potential interventional targets to improve long-term neurological outcomes.

## Figures and Tables

**Figure 1 microorganisms-11-01131-f001:**
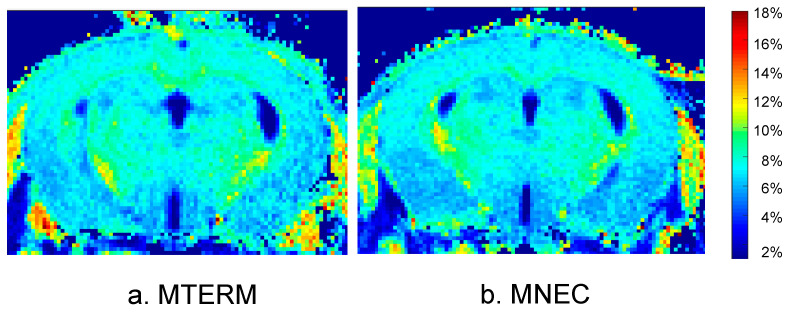
Macromolecular proton fraction (MPF%) map in a four week old mouse in MTERM (**a**) and MNEC (**b**) groups. Color bar indicates pseudo-color mapping of MPF in percent units.

**Figure 2 microorganisms-11-01131-f002:**
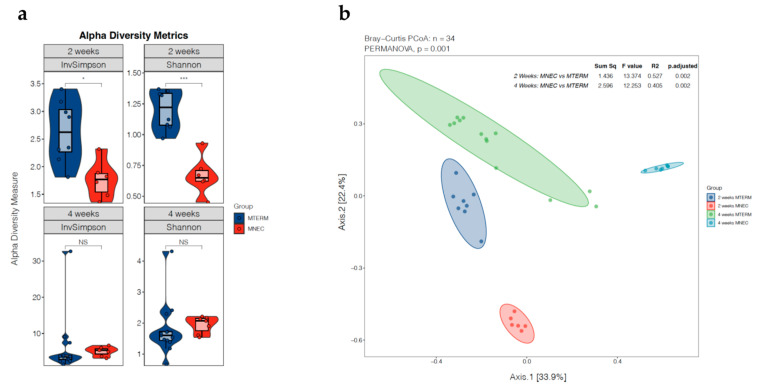
(**a**) α-diversity metrics of lnverse Simpson (lnvSimpson) and Shannon of mouse fecal samples calculated using R package. Inv5impson (* *p* < 0.05) and Shannon (*** *p* < 0.001) of MNEC mouse fecal samples were significantly lower than those of MTERM mouse fecal samples at two weeks of age, but not at four weeks of age (NS, not significant). (**b**) Bray-Curtis principal coordinate analysis (PCoA) scores were plotted based on the relative abundance of fecal microbiota at the genus level. Significant separation in the gut microbiome composition (β-diversity) was observed between MTERM and MNEC by PERMANOVA (overall *p* = 0.001) at two weeks of age (*p* = 0.002) and four weeks of age (*p* = 0.002) while controlling the age group by pair-wise, post hoc tests.

**Figure 3 microorganisms-11-01131-f003:**
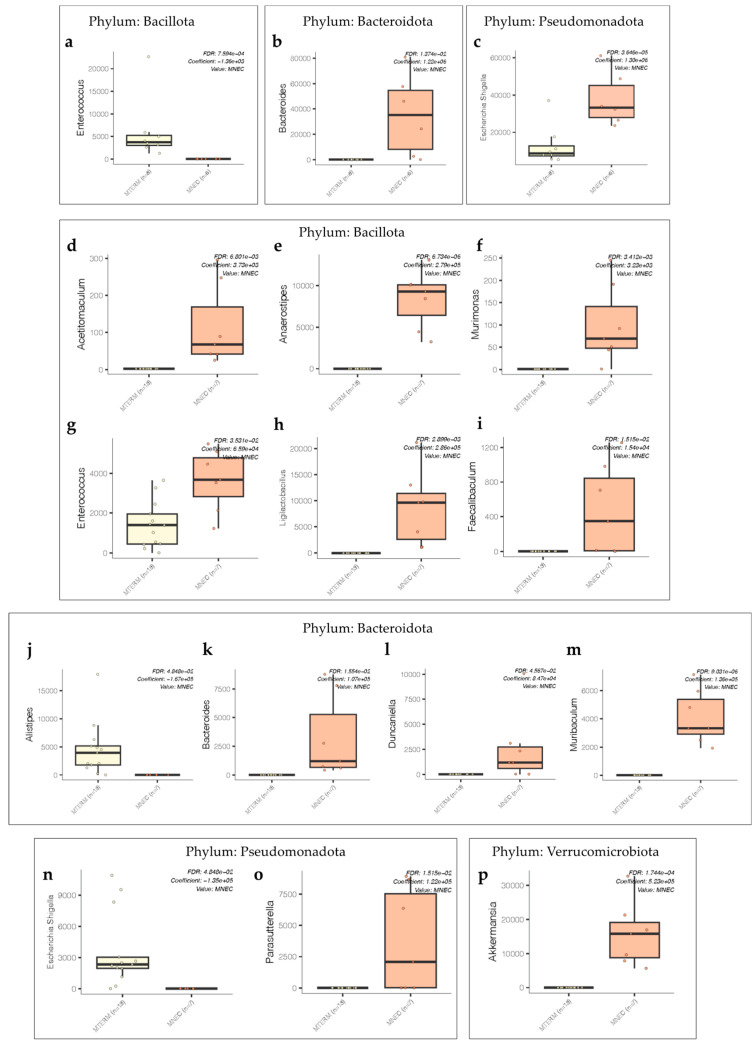
Abundance of bacterial communities between MTERM and MNEC treatment groups. (**a**–**c**) Significant differential abundance at genus level of two weeks old fecal samples. (**d**–**p**) Significant differential abundance at genus level of four weeks old fecal samples. Significant *p* values with FDR adjustment were indicated on the individual figures. Dots represent individual samples.

**Figure 4 microorganisms-11-01131-f004:**
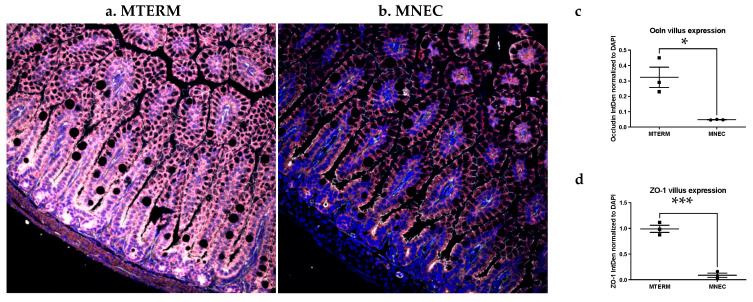
MNEC mice displayed reduced tight junction development in the villus of the ileum at two weeks of age. Representative images of immunofluorescent labeling of Occludin (red) and ZO-1 (white) with DAPI (blue) to counter-staining the nuclei in MTERM (**a**) and MNEC (**b**) mice. There was a significantly decreased Occludin (**c**) and ZO-1 (**d**) in mouse ileal sections in the MNEC (*n* = 3) compared to MTERM (*n* = 3) mice at two weeks of age. ImageJ (NIH) was used to quantify the expression of Occludin and ZO-1 based on integrated density (IntDen). Blocks represent individual samples. Data is presented as mean ± S.E.M. * indicates a significant difference when *p*-value < 0.05 and *** indicates a significant difference when *p*-value < 0.001.

**Figure 5 microorganisms-11-01131-f005:**
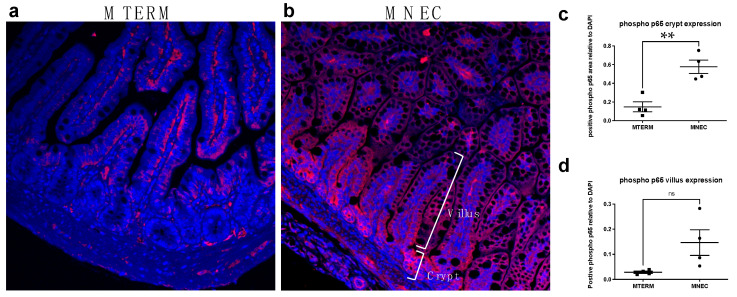
MNEC mice displayed increased crypt inflammation in the ileum at two weeks of age. Representative images of immunofluorescent labeling of phosphor-p65 (red) with DAPI (blue) to counter-staining the nuclei in MTERM (**a**) and MNEC (**b**) mice. There was a significantly increased phosphor-p65 in the crypt (**c**), but not the villus (**d**) in the ileal sections in the MNEC (*n* = 4) compared to those in the MTERM (*n* = 4) mice at two weeks of age. lmageJ (NIH) was used to quantify the co-localization of phosphor-p65 and nuclei. Blocks represent individual samples. Data is presented as mean ± S.E.M. ** indicates a significant difference when *p*-value is <0.01. ns, not significant.

**Figure 6 microorganisms-11-01131-f006:**
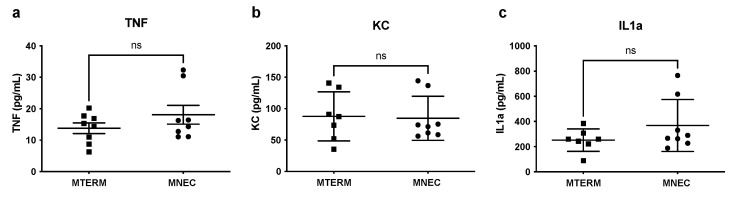
Systemic inflammation was not affected by the two colonized microbial communities. Serum levels of TNF (**a**), KC (**b**), and IL1α (**c**) analyzed by ELISA were not different between MTERM and MNEC groups. Blocks and dots represent individual samples. Data is presented as mean ± S.E.M. Not significant (ns) indicated when *p*-value is >0.05.

**Figure 7 microorganisms-11-01131-f007:**
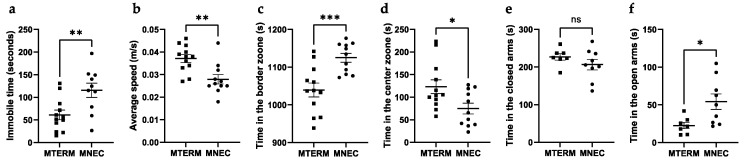
Locomotor and anxiety deficits in MNEC mice. MTERM mice were more mobile (**a**), traveled with faster speed (**b**), spent less time in the border zone (**c**), and spend more time in the center zone (**d**) (*n* = 12 for MTERM mice and *n* = 11 for MNEC mice) in the open field test at four weeks of age. MTERM mice (*n* = 7) were not different from MNEC (*n* = 9) mice in time spent in the closed arms in the elevated plus maze test (**e**). MTERM mice (*n* = 7) spent less time in the open arms than MNEC mice (*n* = 9) (**f**). Blocks and dots represent individual samples. Data is presented as mean ± S.E.M. *, **, and *** indicate significant differences when *p*-value is <0.05, <0.01, and <0.001, respectively. ns, not significant.

**Figure 8 microorganisms-11-01131-f008:**
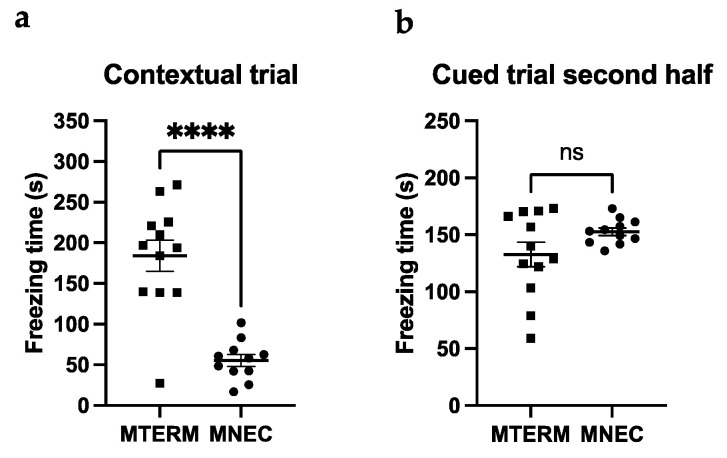
Mioe colonized with fecal samples from NEC infants displayed contextual leaming deficits in contextual/cued fear conditioning tests. (**a**) Freezing time was significantly longer in contextual fear conditioning test (mice were retumed to the same environment of the chamber 24 hours after the oonditioning) in MTERM mice (*n* = 12) at four weeks when compared to the MNEC mice (*n* = 11); (**b**) No differenoe was found in freezing time between MTERM and MNEC mice in the second half of the cued fear conditioning test (different environment of the chamber and sound cue presented) at four weeks. **** indicate a significant difference when *p*-value is <0.0001. ns, not significant. Blocks and dots represent individual samples.

**Figure 9 microorganisms-11-01131-f009:**
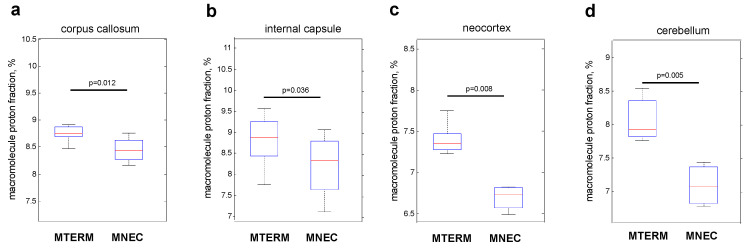
MPF was significantly decreased in major white matter tracts (corpus callosum (**a**) and internal capsule (**b**)), cerebral cortex (**c**), and cerebellum (**d**) in MNEC relative to MTERM group. Animal numbers at four weeks of age (*n* = 5). *p*-value < 0.05 is considered significant different between the groups.

**Figure 10 microorganisms-11-01131-f010:**
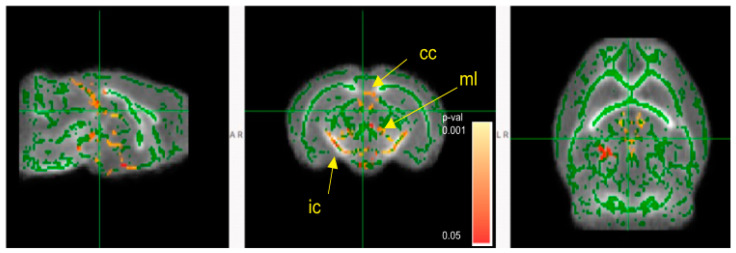
Decreased fractional anisotropy (FA) in MNEC mice relative to MTERM mice based on TBSS analysis. Green skeletonized white matter image was overlaid on mean group FA map, shown on sagittal coronal and horizontal sections. Red-orange pseudo-colored statistical parametric map of *t*-test *p*-values represent voxels where FA was significantly higher in MTERM mice (*p* corrected < 0.05). cc—corpus callosum, ic—internal capsule, ml—medial lemniscus. Animal numbers at four weeks of age (*n* = 5).

**Figure 11 microorganisms-11-01131-f011:**
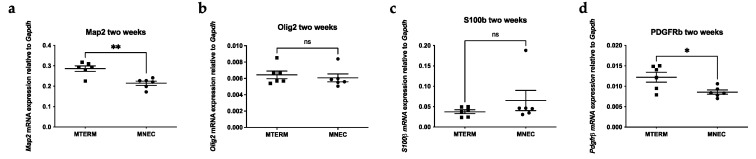
Relative brain transcripts of *Map2* ((**a**), *n* = 6 for both groups at two weeks of age), *Olig2* (**b**), *S100b* (**c**), and *Pdgfr* (**d**) were measured by RT-PCR. Data is presented as mean ± S.E.M. and normalized to *Gapdh* mRNA transcript levels, Blocks and dots represent individual samples. ns, not significant. * and ** indicate significant differences when *p*-value is <0.05 and <0.01, respectively.

**Figure 12 microorganisms-11-01131-f012:**
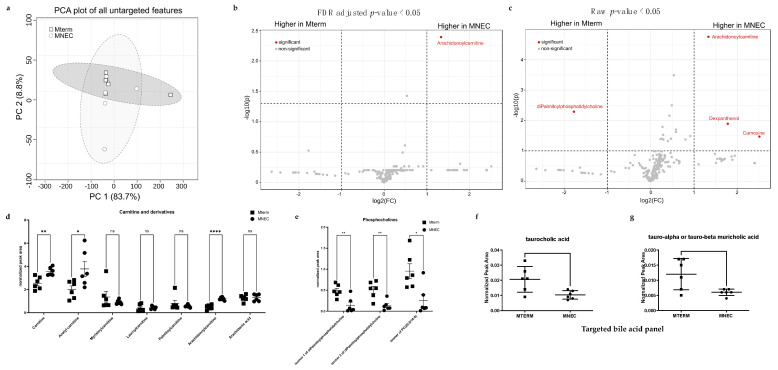
Brain metabolic profiles of MTERM and MNEC mice (**a**) principal component analysis (PCA) plot showed there was no separation between the two groups (both *n* = 6). (**b**) Volcano plot showing relative abundance of putatively identified features according to GNPS with FDR-adjusted *p* values. (**c**) Volcano plot showing relative abundance of putatively identified features according to GNPS without FDR-adjusted *p* values. In both (**b**,**c**) plots, the X-axis represents the abundance fold change on log2 scale, and the Y-axis represents the negative log10 of the calculated *p* value. (**d**) Reltive abundance of the analyzed features with putative IDs of carnitine analogs between the two groups. (**e**) Relative abundance of the analyzed features with putative IDs of phosphocholine analogs between the two groups. Targeted metabolic profiling revealed relative abundance levels of (**f**) Taurocholic and (**g**) tauro-alpha or tauro-betamuricholic arid from the bile acid panel were significantly lower in the brain samples of NEC mice. MetaboAnalyst was used to generate the plots and statistics. *, **, and **** indicate significant differences when *p*-value is <0.05, <0.01, and <0.0001, respectively. ns, not significant. Blocks and dots represent individual samples.

## Data Availability

The data generated from this study are available upon request from the corresponding author.

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
