# Peer review of "Microbiota from Preterm Infants Who Develop Necrotizing Enterocolitis Drives the Neurodevelopment Impairment in a Humanized Mouse Model"

_microorganisms, 2023, doi:10.3390/microorganisms11051131_

Round 1
Reviewer 1 Report
Manuscript ID: microorganisms-2347046
Type of manuscript: Article
Title: Microbiota from preterm infants who develop necrotizing enterocolitis drives the neurodevelopment impairment
This study tested the hypothesis that microbial communities prior to the onset of necrotizing enterocolitis (NEC) drive neurodevelopment impairment (NDI). Using a humanized gnotobiotic model in which microbial samples from human infants were gavaged into germ-free C57BL/6J pregnant mothers, the effects of the microbiota of preterm infants who developed NEC (MNEC) were compared with the microbiota of healthy infants at term (MTERM) on brain development and neurological outcomes in mouse pups.
Comments and Suggestions for Authors:
The manuscript is a very interesting study, but requires some considerations.
The title should make it clear that the experimentation has been carried out in a mouse model. Could be changed by eg Microbiota from preterm infants who develop necrotizing enterocolitis drives the neurodevelopment impairment in off-spring mice.
In this study the authors did not detect significant difference in serum proinflammatory mediators between the MTERM and MNEC groups. They carry out an adequate discussion of the matter, but in the possibility "3) both groups of mice were breast fed by the dams which would not happen in most of the preterm infants or NEC patients", more reflection should be introduced in this regard due to the possibility of its practical application. Appropriate bibliographical references should be provided.
The conclusions should be modulated to the fact that it is animal experimentation. Where it says "we have demonstrated that microbiota from human preterm infants who went on to develop NEC drives deficits in brain development and neurological outcomes" should be added in off-spring mice.
References should be thoroughly revised to conform to uniform and appropriate standards for the journal Microorganisms.
Minor editing of English language required
Author Response
The manuscript is a very interesting study but requires some considerations.
The title should make it clear that the experimentation has been carried out in a mouse model. Could be changed by eg Microbiota from preterm infants who develop necrotizing enterocolitis drives the neurodevelopment impairment in off-spring mice.
We have changed the title to “Microbiota from preterm infants who develop necrotizing enterocolitis drives the neurodevelopment impairment in a humanized mouse model.”
In this study the authors did not detect significant difference in serum proinflammatory mediators between the MTERM and MNEC groups. They carry out an adequate discussion of the matter, but in the possibility "3) both groups of mice were breast fed by the dams which would not happen in most of the preterm infants or NEC patients", more reflection should be introduced in this regard due to the possibility of its practical application. Appropriate bibliographical references should be provided.
We agree with the reviewer that preterm infants fed with mother’s milk have reduced risk for neonatal infection. We have included additional information and references shown below in the Discussion.
“In human studies, breastfeeding or exclusive human milk diet is associated with reduced risk for neonatal infection and NEC90-92 . It is important to point out that many mothers with preterm delivery often
have difficulty providing adequate amounts of breast milk or breast milk with adequate nutritional contents90 . In fact, nutritional content in preterm infants’ diet is a key regulator of neurodevelopment outcome independent of neonatal infection90 , suggesting that modifiable factors such as diet or microbiota as we identified in this study might collectively contribute to the neurodevelopment outcomes of preterm infants”.
The conclusions should be modulated to the fact that it is animal experimentation. Where it says "we have demonstrated that microbiota from human preterm infants who went on to develop NEC drives deficits in brain development and neurological outcomes" should be added in off-spring mice.
We have added “in the offspring in a humanized mouse model.” in the Conclusion.
References should be thoroughly revised to conform to uniform and appropriate standards for the journal Microorganisms.
References have been changed to the appropriate style of Microorganisms.
Reviewer 2 Report
The study examines the effect of microbial communities on brain development and neurological outcomes using a mouse model. Microbiota from preterm infants who went on to develop necrotizing enterocolitis (NEC) were compared to microbiota from healthy term infants. The results showed that microbial communities from infants who developed NEC negatively affected ileal barrier development and homeostasis, as well as causing delayed brain maturation and organization. Mice with MNEC had worse mobility, anxiety, and contextual memory than those with term infants (MTERM). The study suggests that the microbiome before the onset of NEC has negative impacts on brain development and neurological outcomes. This might be an interesting study. However, I have several major concerns.
1. Line 127. Provide the full name of ASVs.
2. In Section 2.7, it may not be easy to understand the behavioral tests by reading this paper. Are there any pictures that can help readers understand these tests?
3. Figure 1 only shows the results of two mice. Is it different significantly for other mice? Why does Figure 1 present in the Method section, it should be presented in the result section?
4. Figures 2 and 3 are too small. It is hard to read the words.
5. Lines 362. “Samples from MTERM (n=8) and MNEC (n=6) pups at two weeks of age and MTERM (n=13) and MNEC (n=7) pups at four weeks of age were subjected for 16S rRNA sequencing.” Why are the sample sizes at two weeks of age and at four weeks of age different? If you use the same sample size (MTERM (n=8) and MNEC (n=6)) for the four weeks of age, is it still significantly different? Do you have the result for the six weeks of age?
6. Based on the result presented in Figure 2, there was no significant difference in Alpha diversity between the MTERM and MNEC groups after four weeks, but there was a significant difference for the two weeks period. This might suggest that NEC may not have long-term negative effects, and that the body has the ability to recover on its own. This may contradict the result of this study.
7. The font used in Legend of Figure 12 is different from that of the other figures
Author Response
Comments and Suggestions for Authors
The study examines the effect of microbial communities on brain development and neurological outcomes using a mouse model. Microbiota from preterm infants who went on to develop necrotizing enterocolitis (NEC) were compared to microbiota from healthy term infants. The results showed that microbial communities from infants who developed NEC negatively affected ileal barrier development and homeostasis, as well as causing delayed brain maturation and organization. Mice with MNEC had worse mobility, anxiety, and contextual memory than those with term infants (MTERM). The study suggests that
the microbiome before the onset of NEC has negative impacts on brain development and neurological outcomes. This might be an interesting study. However, I have several major concerns.
1. Line 127. Provide the full name of ASVs.
“ASVs” has been spelled out as “amplicon sequence variants”.
2. In Section 2.7, it may not be easy to understand the behavioral tests by reading this paper. Are there any pictures that can help readers understand these tests?
The behavioral tests we used in this study, open field, elevated plus maze and contextual and cued fear conditioning tests, are standard assessments used in mouse behavior studies. We have provided the reference in Methods and described the procedures in each testing protocol.
3. Figure 1 only shows the results of two mice. Is it different significantly for other mice? Why does Figure 1 present in the Method section, it should be presented in the result section?
Figure 1 in Methods with the images (one from each our treatment group) illustrated how macromolecule proton fraction (MPF) was obtained from MRI. In Results 3.5, the quantification of the means of MPF of each treatment group and statistics (n=5) are demonstrated in Figure 9.
4. Figures 2 and 3 are too small. It is hard to read the words.
We have enlarged Figure 2 and Figure 3.
5. Lines 362. “Samples from MTERM (n=8) and MNEC (n=6) pups at two weeks of age and MTERM (n=13) and MNEC (n=7) pups at four weeks of age were subjected for 16S rRNA sequencing.” Why are the sample sizes at two weeks of age and at four weeks of age different? If you use the same sample size (MTERM (n=8) and MNEC (n=6)) for the four weeks of age, is it still significantly different? Do you have the result for the six weeks of age?
Based on our previous studies and power analysis, the minimum number required to reach the significance at p<0.05 is six. We intentionally collected the pups from at least three litters to take the consideration of litter effect. Each litter has a different litter size (for C57/BL6 the litter size is in the range of four to 10), therefore we had different n in each treatment and time point. The study was terminated at four weeks of age. We do not have samples at six weeks of age for analysis.
6. Based on the result presented in Figure 2, there was no significant difference in Alpha diversity between the MTERM and MNEC groups after four weeks, but there was a significant difference for the two weeks period. This might suggest that NEC may not have long-term negative effects, and that the body has the ability to recover on its own. This may contradict the result of this study.
Our early study demonstrated that the intestinal microbiome samples from preterm infants up through five weeks of life cluster distinctly from those of a full term breast-fed infant, and that the microbial patterns converge toward those of full term breast-fed infants only at or after six weeks of life 1 . These observations were confirmed by others demonstrating that preterm infant microbiota displayed lower diversity and more Proteobacteria and Enterococcus compared with full-term infants at 10 days postpartum2. However, the difference between preterm and term infants in bacteria diversity was not observed at four months or 12 months postpartum, consistent with our study observing a shift of preterm microbiome to full term infant microbiome patterns at six weeks of age and later. Our results display the importance of early colonization and development of the preterm infants’ microbiome on later brain development and agree with the notion that targeting the critical window of initial colonization and early development of gut microbiome can have impacts on neurodevelopment.
7. The font used in Legend of Figure 12 is different from that of the other figures The font used in Figure 12 has been changed.
(1) Claud, E. C.; Keegan, K. P.; Brulc, J. M.; Lu, L.; Bartels, D.; Glass, E.; Chang, E. B.; Meyer, F.; Antonopoulos, D. A. Bacterial community structure and functional contributions to emergence of health or necrotizing enterocolitis in preterm infants. Microbiome 2013, 1 (1), 20. DOI: 10.1186/2049-2618-1-20.
(2) Dahl, C.; Stigum, H.; Valeur, J.; Iszatt, N.; Lenters, V.; Peddada, S.; Bjornholt, J. V.; Midtvedt, T.; Mandal, S.; Eggesbo, M. Preterm infants have distinct microbiomes not explained by mode of delivery, breastfeeding duration or antibiotic exposure. Int J Epidemiol 2018. DOI: 10.1093/ije/dyy064.
Round 2
Reviewer 2 Report
The authors have addressed some of my comments.